# Quantitative Proteomics Reveals Fh15 as an Antagonist of TLR4 Downregulating the Activation of NF-κB, Inducible Nitric Oxide, Phagosome Signaling Pathways, and Oxidative Stress of LPS-Stimulated Macrophages

**DOI:** 10.3390/ijms26146914

**Published:** 2025-07-18

**Authors:** Albersy Armina-Rodriguez, Bianca N. Valdés Fernandez, Carlimar Ocasio-Malavé, Yadira M. Cantres Rosario, Kelvin Carrasquillo Carrión, Loyda M. Meléndez, Abiel Roche Lima, Eduardo L. Tosado Rodriguez, Ana M. Espino

**Affiliations:** 1Department of Microbiology and Medical Zoology, University of Puerto Rico-Medical Sciences Campus, San Juan, PR 00936, USA; albersy.armina@upr.edu (A.A.-R.); bianca.valdes@upr.edu (B.N.V.F.); carlimar.ocasio@upr.edu (C.O.-M.); loyda.melendez@upr.edu (L.M.M.); 2Translational Proteomics Center, Research Capacity Core, Center for Collaborative Research in Health Disparities (CCRHD), Academic Affairs Deanship, University of Puerto Rico-Medical Sciences Campus, San Juan, PR 00936, USA; yadira.cantres@upr.edu; 3Integrated Informatics Services, Research Capacity Core, Center for Collaborative Research in Health Disparities (CCRHD), Academic Affairs Deanship, University of Puerto Rico-Medical Sciences Campus, San Juan, PR 00936, USA; kelvin.carrasquillo@upr.edu (K.C.C.); abiel.roche@upr.edu (A.R.L.); or tosadoe1@uagm.edu (E.L.T.R.); 4School of Dental Medicine, Universidad Ana G. Mendez, Gurabo, PR 00778, USA

**Keywords:** TMT-labeling, proteomics, *Fasciola hepatica*, Fh15, TLR, NF-κB, iNOS, CD36, Lck, LPS, macrophages, sepsis

## Abstract

There is a present need to develop alternative biotherapeutic drugs to mitigate the exacerbated inflammatory immune responses characteristic of sepsis. The potent endotoxin lipopolysaccharide (LPS), a major component of Gram-negative bacterial outer membrane, activates the immune system via Toll-like receptor 4 (TLR4), triggering macrophages and a persistent cascade of inflammatory mediators. Our previous studies have demonstrated that Fh15, a recombinant member of the *Fasciola hepatica* fatty acid binding protein family, can significantly increase the survival rate by suppressing many inflammatory mediators induced by LPS in a septic shock mouse model. Although Fh15 has been proposed as a TLR4 antagonist, the specific mechanisms underlying its immunomodulatory effect remained unclear. In the present study, we employed a quantitative proteomics approach using tandem mass tag (TMT) followed by LC-MS/MS analysis to identify and quantify differentially expressed proteins that participate in signaling pathways downstream TLR4 of macrophages, which can be dysregulated by Fh15. Data are available via ProteomeXchange with identifier PXD065520. Based on significant fold change (FC) cut-off of 1.5 and *p*-value ≤ 0.05 criteria, we focused our attention to 114 proteins that were upregulated by LPS and downregulated by Fh15. From these proteins, TNFα, IL-1α, Lck, NOS2, SOD2 and CD36 were selected for validation by Western blot on murine bone marrow-derived macrophages due to their relevant roles in the NF-κB, iNOS, oxidative stress, and phagosome signaling pathways, which are closely associated with sepsis pathogenesis. These results suggest that Fh15 exerts a broad spectrum of action by simultaneously targeting multiple downstream pathways activated by TLR4, thereby modulating various aspects of the inflammatory responses during sepsis.

## 1. Introduction

Sepsis remains one of the leading causes of mortality among surgical patients and individuals in intensive care units worldwide [1]. Although antibiotic therapy has improved patient prognosis, the neutralization of the inflammatory response is essential to prevent the progression of clinical symptoms and reduction in mortality associated with a persistent inflammatory cascade.

Helminths have developed multiple strategies to modulate the host immune system and establish long-term chronic infections. As immunomodulatory organisms, they predominantly induce strong Th2/Treg immune responses [2,3]. It is believed that this strategy is mutually beneficial, to the host and the parasite, as it protects the host from the serious consequences of inflammation, while the extermination of the worm is hindered [4]. It has been speculated that such sophisticated immunoregulatory capacity results from hundreds of millions of years of co-evolution with humans, leading to an immune system adapted to these organisms [5,6]. According to the ‘old-friends hypothesis’, the removal of helminths and their anti-inflammatory influence, from our environment may partly explain the emergence of immunological disorders in western countries [7,8]. Studies have also demonstrated that concurrent helminth infections can counterbalance the exacerbated bacteria-induced pro-inflammatory responses seen during the phases of sepsis known as SIRS (Systemic Inflammatory Response Syndrome) and CARS (Compensatory Anti-inflammatory Response Syndrome), thereby improving survival [1]. Moreover, research using animal models, epidemiological studies as well as clinical trials suggests that both natural and artificial helminth infections could protect against various immunological disorders [9,10,11,12,13,14,15]. However, although the helminth-therapy has gained significant credibility, it is not universally accepted by the scientific community. There are still numerous ethical controversies associated with the helminth-therapy, and the inability to effectively control the course of infection raises major safety concern. In addition, the most immunomodulatory helminth species such as *Schistosoma, Fasciola* and *Clonorchis*, among others, are highly pathogenic, and therefore, contraindicated for use as helminth-therapy, which makes this approach impractical. In this sense, the identification of well-defined helminth molecules that ultimately mediate the host-immunomodulation could serve as a template for the design of novel anti-inflammatory drugs.

Our research group has demonstrated that members of the *Fasciola hepatica* fatty acid-binding proteins (FABPs), particularly Fh15, is one of the most immunomodulatory molecules with potential to develop biotherapeutic drugs against inflammatory disorders. Using experimental models of sepsis, including mice and non-human primates, treatment with Fh15 effectively suppressed inflammatory markers associated with Th1-responses with excellent tolerability and no apparent side effects [16,17,18,19]. Consequently, Fh15 has been proposed as a TLR4 antagonist, leveraging its remarkable capacity to prevent or treat hyperinflammation induced by LPS or live *Escherichia coli* [18,19]. Despite being one of the most extensively studied helminth-derived molecules, the precise mechanisms by which Fh15 exerts its immunomodulatory effects remain unknown.

It has been hypothesized that Fh15 may downregulate key proteins downstream TLR4 in macrophages in the same way as Fh12, the native variant of FABP [17], which comprises a mixture of at least eight isoforms [20]. Macrophages are critical immune cells involved in the development and progression of sepsis, particularly in response to bacterial infections. They express TLR4 on their surface, a key receptor for recognizing and responding to LPS from Gram-negative bacteria. During early phases of sepsis, macrophages become excessively activated and polarized toward M1, releasing a pro-inflammatory cytokine storm. This response trigger coagulation factors and subsequent physiological disturbances that can lead to organ failure and, ultimately, death [1].

Herein, we applied the TMT-based quantitative proteomics approach to identify and quantify proteins differentially expressed in LPS-treated RAW 264.7 macrophage cells, focusing on those downregulated by Fh15. This method also allowed us to elucidate the biological processes and signaling pathways in which these proteins participate. RAW 264.7 cells have been extensively used in numerous proteomic studies [21,22,23]. This approach will enable us to build a more complete picture of how Fh15 exerts its anti-inflammatory effect and will set a foundation for the development of a novel class of biotherapeutic agents.

## 2. Results

### 2.1. Quantitative Proteomics Analysis of Macrophages-like Cells Exposed to LPS or Fh15

Total proteins from macrophage-like cells (RAW 264.7 cells), treated in triplicate with LPS, Fh15 or Phosphate-buffered saline (PBS) were separately analyzed by LC-MS/MS for protein identification (Figure 1). A total of 10,943 proteins were identified in LPS-treated samples, while 10,934 were identified in Fh15-treated samples. Bioinformatics analyses revealed that the expression of most of the differentially abundant proteins was increased in LPS-stimulated macrophages when compared to PBS control cells. Conversely, when comparing Fh15-treated cells to LPS-treated cells as the control, these proteins showed decreased levels of abundance, as illustrated in the volcano plots (Figure 2A,B).

To ensure the bioinformatics analysis focused on a set of proteins with statistically significant abundance, we defined as dysregulated those proteins exhibiting a Log2 FC ≥ |0.5| with a *p*-value ≤ 0.05, where upregulated are the proteins with the FC value greater or equal to than 0.5 (Log2 FC ≥ 0.5), and downregulated are the proteins with FC values less than or equal to −0.5 (Log2 FC ≤ −0.5). According to this definition, we identified and quantified 280 dysregulated proteins (185 upregulated and 95 downregulated) in samples stimulated with LPS compared to PBS treatment, and 150 dysregulated proteins (111 upregulated and 39 downregulated) in samples treated with Fh15 compared to LPS (Figure 3A). Among these, a total of 114 dysregulated proteins were common to both treatments (83 upregulated and 31 downregulated in response to LPS and 31 upregulated and 83 downregulated in response to Fh15), which represented 40.7% of all dysregulated proteins by LPS and 76.0% of the dysregulated proteins by Fh15 (Appendix A). Excluding the common proteins, we found that 166 dysregulated proteins (59.3%) were unique to macrophage-like stimulated with LPS (102 upregulated and 64 downregulated) while 36 dysregulated proteins (24%) were unique for macrophages-like treated with Fh15 (8 upregulated and 28 downregulated) (Figure 3B). To visualize the expression patterns, a heat map was generated for the 114 common proteins, between both LPS and Fh15 treatments, ordered by the magnitude of their fold change (Figure 4, Appendix A).

### 2.2. Subcellular Localization and Function of Dysregulated Proteins

Understanding the subcellular localization of proteins is essential, as it plays a pivotal role in determining their functions. To explore this aspect, we used Ingenuity Pathway Analysis (IPA) to predict the subcellular localization of the 114 proteins common to both comparisons. The analysis revealed that most of these dysregulated proteins (60 proteins) are localized in the cytoplasm, followed by the plasma membrane (24 proteins), the nucleus (17 proteins), the extracellular space (10 proteins), and other cellular compartments (3 proteins) (Figure 5A). In terms of the functional categorization of the common dysregulated proteins, enzymes emerged as the most prevalent group, comprising a total of 27 proteins. Within this category, kinases (12 proteins), phosphatases (3 proteins), and peptidases (3 proteins) were particularly notable. Additionally, dysregulated proteins involved in transport (8 proteins), transcription regulation (9 proteins), and transmembrane signaling (6 proteins, including 5 transmembrane receptors and 1 G-protein coupled receptor) were also prominent. The presence of cytokines (5 proteins) and ion channels (4 proteins) further highlights the diverse functional roles influenced by both treatments (Figure 5B). This comprehensive analysis underscores the importance of subcellular localization in understanding protein function and emphasizes the intricate regulatory networks activated in response to LPS and Fh15 treatment. Notably, a category labeled as “Other” included 36 proteins, indicating a range of additional functions yet to be fully explored.

### 2.3. IPA Results and Functional Enrichment Analysis

We performed an IPA of the 114 dysregulated proteins that were found to be common between the two comparison groups (LPS vs. PBS and Fh15 vs. LPS) to identify which proteins participate in the NF-κB, iNOS, acute phase response and phagosome signaling pathways, and are closely associated with sepsis pathogenesis [25,26,27,28]. We focused our attention on six proteins with high fold change expression values participating in these canonical signaling pathways (Table 1). The proteins were: interleukin-1 alpha (IL-1α) (FC = 2.195, *p* = 0.006) and tumor necrosis factor-alpha (TNF-α) (FC =1.597, *p* = 0.004), which are potent inflammatory cytokines that strongly activate NF-κB, a key regulator of inflammation [29,30], lymphocyte specific-protein tyrosine protein kinase (Lck) (FC = 2.736, *p* = 0.006), which along with NF-κB are crucial players in B-cell and T cell activation and signaling [31,32] (Appendix A), inducible nitric oxide synthase-2 (NOS2) (FC = 4.584, *p* = 0.00001), which play key roles in the production of nitric oxide and the anti-microbial activity of macrophages [33] (Appendix A), manganese superoxide dismutase-2 (SOD2) (FC = 1.722, *p* = 0.00008) related to the acute phase response and oxidative stress during inflammation and injury [34] (Appendix A), and cluster of differentiation-36 (CD36) (FC = 2.038, *p* = 0.0008) involved in bacterial phagocytosis by macrophages [35] (Appendix A). Conversely, all these proteins were found downregulated in macrophages treated with Fh15. IL-1α and TNF-α exhibited a fold change of −1.79 (*p* = 0.04 and *p* = 0.005, respectively), NOS2 showed a FC = −4.24 (*p* = 0.00003), Lck was found to decrease by −2.14 (*p* = 0.01), whereas SOD2 and CD36 were found to decrease by −1.67 (*p* = 0.0005) and −1.54 (*p* = 0.0006), respectively.

### 2.4. Validation of Selected Downregulated Proteins by Fh15 Using Western Blot

We selected Lck, IL-1α, TNF-α, NOS2, CD36 and SOD2 for validation by Western blot based on their statistical significance as downregulated proteins and their roles in several inflammatory pathways. For this, BMDMs were treated with Fh15 for 30 min and then stimulated overnight with LPS. BMDMs treated only with Fh15 and PBS or stimulated with LPS were used as controls. Densitometry analysis quantified protein expression levels relative to PBS-treated controls, calculating fold changes across different treatment groups (BMDMs treated with LPS, Fh15 or Fh15 + LPS). The results revealed a downregulation of all six proteins in samples treated with Fh15 or Fh15 + LPS. Conversely, in samples treated only with LPS; these proteins were upregulated (Figure 6). Specifically, Lck, CD36, IL-1α, SOD2 and NOS2 were between 1.5- and 1.9-fold more expressed in BMDMs stimulated with LPS than PBS-control. In contrast, the relative amount of Lck, CD36, IL-1α and NOS2 in BMDMs treated with Fh15 was like the PBS control. SOD2 was the only validated protein that showed an increase of 1.3-fold higher than the PBS-control, which was found significant (*p* = 0.0143). Importantly, in BMDMs treated with Fh15 + LPS all proteins, including SOD2 were remarkably reduced when compared to LPS-stimulated cells (Lck *** *p* = 0.0005, CD36 ** *p* = 0.008, IL-1α *** *p* = 0.0003, NOS2 * *p* = 0.05, SOD2 * *p* = 0.04, and TNF-α ** *p* = 0.008). A similar pattern was observed when validation was performed with samples generated from RAW 264.7 cells treated with LPS or Fh15 (Appendix A).

### 2.5. Measurement of TNF-α Levels in Supernatant of Culture from Bone Marrow-Derived Macrophages (BMDMs)

Since Fh15, as a TLR4-mediated immunoregulator, may influence the secretion of key pro-inflammatory cytokines, we measure TNF-α levels in the culture supernatant of BMDMs treated with Fh15, LPS or a combination of Fh15 and LPS. Our results showed that TNF-α levels in the supernatant of BMDMs treated Fh15 were nearly undetectable, similar to those in the PBS-treated control cells. In contrast, supernatants from BMDMs stimulated with LPS alone exhibited an average TNF-α concentration of 8.833 ± 0.257 pg/mL. Notably, the secreted TNF-α levels were significantly reduced (*p* = 0.0001) when cells were treated first with Fh15 and then stimulated with LPS (Figure 7).

## 3. Discussion

In sepsis, the severe systemic inflammatory response, known as cytokine storm, involves the excessive release of inflammatory cytokines such as TNF-α and IL-1α into the bloodstream. This phenomenon is primarily triggered by the activation of pattern recognition receptors (PPRs), particularly toll-like receptor-4 (TLR4) on macrophages and other innate immune cells, which recognize and bind LPS, a potent endotoxin of Gram-negative bacteria [36]. This engagement triggers signaling pathways that lead to the activation of transcription factors, in which NF-κB is central [37,38]. Activation of NF-κB promotes the transcription of multiple pro-inflammatory genes, resulting in excessive inflammation, organ dysfunction and increased mortality [26,27]. Our previous studies with animal models, including mice and non-human primates, demonstrated that a single dose of Fh15 administered prior to or after exposure to lethal doses of LPS or live *E. coli*, respectively, was sufficient to suppress the cytokine storm and several other inflammatory markers [18,19]. Consistent with this, our current proteomics analysis revealed that TNF-α and IL-1α were found significantly downregulated in macrophages treated with Fh15 followed by LPS. This result also is consistent with previous in vitro observations using a monocyte cell line (THP1-Blue CD14 cells) in which Fh15 significantly suppressed the NF-κB activation in response to various Gram-negative and Gram-positive bacteria strains [39]. Blocking NF-κB represents a feasible strategy for treating inflammatory conditions like sepsis, given its central role in disease pathogenesis [26,27]. However, complete blockage of NF-κB can impair other functions, such as essential host defenses and tissue homeostasis, potentially leading to adverse effects. In that sense, we consider Fh15 a specific NF-κB inhibitor that is not toxic for the cell and does not compromise essential cellular functions [18,19,39].

Another notably finding of this study was the observation that Fh15 downregulates two NOS variants. The iNOS and NF-κB pathways are closely interconnected during inflammation caused by sepsis, and their dysregulation can contribute significantly to tissue damage. Activation of NF-κB promotes the upregulation of inducible nitric oxide synthase (iNOS), the enzyme responsible for catalyzing the production of large amount of nitric oxide (NO) from arginine during inflammation [40]. Consequently, the interplay between NF-κB and NOS creates a complex feedback loop where the activation of NF-κB induces iNOS and NO production, and both in turn can modulate NF-κB [41]. Considering that during the early phase of sepsis, macrophages undergo M1 polarization, characterized by increased expression of iNOS and TNF-α as well as the production of nitrogen species, the observation that Fh15 downregulates two NOS variants and suppress the expression and secretion of TNF-α indicates that Fh15 can concurrently modulate both the iNOS and NF-κB pathways. It is well established that downregulation of iNOS shifts macrophage polarization towards M2 phenotype (alternative activated macrophages), which is associated with anti-inflammatory functions, tissue repair, wound healing, and the resolution of the inflammation [42,43,44]. However, among the 114 common proteins identified by our proteomics analysis, we do not detect any protein associated with M2 that were upregulated by Fh15. Given that RAW 264.7 cells are a well-established model for studying M2 polarization [45,46] and considering the TMT-labeling combined with LC/MS-MS offers a highly sensitive quantitative approach [47], the failure in detecting M2-associated proteins may indicate that Fh15 does not directly influence macrophage polarization toward M2. Interestingly, M2 polarization is a hallmark of helminth infections [48,49,50], including those caused by *F. hepatica* [51,52,53]. Moreover, *F. hepatica* tegument antigens and excretory-secretory products, including fatty acid binding protein (FABP) among their components [54,55], have been shown to indirectly induce an M2-like macrophage phenotype in vivo [51,56,57]. Additionally, native FABP (Fh12), a complex of at least eight isoforms from which Fh15 is one of them [20], has shown to induce human M2-macrophage in vitro [58]. Therefore, although the observation that Fh12 induces M2 phenotype and Fh15 does not might seem contradictory, it is plausible that the role of Fh15 is only to suppress the M1 inflammatory phenotype, whereas the ability to influence the M2 polarization may rely on other FABP isoforms or *F. hepatica* antigens.

The observation that Fh15 downregulates SOD2 represent another compelling finding. The NF-κB pathway is closely linked to oxidative stress, a hallmark that characterize the pathogenesis of sepsis [59]. During sustained inflammatory responses, mitochondrial oxygen consumption becomes impaired, leading to excessive production of superoxide radicals, highly reactive molecules capable of damaging cellular components [60]. Normally, superoxide radicals are converted into hydrogen peroxide (H_2_O_2_) by antioxidant enzymes such as superoxide dismutase 2 (SOD2), and subsequently H_2_O_2_ is detoxified to H_2_O by other enzymes such as catalase (CAT). Therefore, SOD2 plays a crucial protective role against oxidative damage; however, alteration in the SOD:CAT ratio, for example, an overexpression of SOD2, could lead to an increase in the levels of oxidative stress and morbidity in sepsis [61]. Consistent with these premises, the high levels of SOD2 observed in RAW 264.7 cells and BMDMs stimulated with LPS could be an indicative of oxidative stress. On the other hand, the high levels of iNOS induced by LPS in these cells suggest overproduction of nitric oxide (NO), which can react with superoxide to generate of reactive oxygen species (ROS), further amplify oxidative damage. These events contribute to an imbalance between the production of ROS and cellular capacity to detoxify them, ultimately leading to cell damage [62,63]. Importantly, our data show that BMDMs treated with Fh15 exhibit significantly higher levels of SOD2 compared to PBS-treated control cells, while co-treatment with Fh15 and LPS (Fh15 + LPS) results in a marked reduction in SOD2 levels relative to LPS alone. This suggests that Fh15 can reduce the excess of SOD2 induced by LPS and bring it back to healthy levels as a mechanism to protect cells from oxidative stress damage.

Another important finding of our study is that Fh15 downregulates Lck (lymphocyte-specific protein tyrosine kinase), a critical regulator involved in NF-κB activation [64]. This protein also plays a crucial role in T cell activation [65] and is essential for the development of effector CD4 and CD8 lymphocytes as well as memory T cells [66]. Abnormal expression of Lck and NF-κB have been associated with various autoimmune diseases and malignancies, including systemic lupus erythematosus (SLE) [67], rheumatoid arthritis (RA) [68], acute T cell lymphocyte leukemia (T-ALL) [69] and cholangiocarcinoma [70]. Notably, Lck is currently considered a potential biomarker for sepsis [71] due to its elevated levels frequently observed in septic patients [72]. In the context of helminth infections, Lck has also been identified as an immunomodulator of the immune system by promoting the T-helper-2 immune response characterized by the secretion IL-4, IL-5 and IL-13 [73]. Thus, being Fh15 is a helminth-derived molecule, its ability to downregulate Lck may suggest an anti-inflammatory mechanism based in promoting Th2-responses. Importantly, studies have demonstrated that Lck interact with CD36 to modulate various cellular processes, including T cell activation and immune responses [74]. CD36, a scavenger receptor localized on the cell surface, participates in several cell functions including lipid metabolism [75] and collaborate with TLR4 in the LPS recognition during early stages of infection [76]. CD36 also play a role in the internalization of both Gram-negative and Gram-positive bacteria [77], facilitating phagocytosis and promoting cytokine secretion from pathogen-stimulated cells [76,78]. In sepsis lipid metabolism is significantly dysregulated and this dysregulation has been associated with elevated levels of CD36, which contribute with the worsening of sepsis pathology, organ dysfunction and increased mortality [79]. Given its involvement in sepsis pathogenesis, CD36 is currently considered an important target for sepsis treatment since the suppression of CD36 has been associated with ameliorated symptoms and improve survival outcomes [80,81]. Therefore, the observation that Fh15 significantly downregulate CD36 represents one of the most relevant findings of our study and reinforce the potential of Fh15 as biotherapeutic against sepsis.

Although we selected for validation six proteins downregulated by Fh15 that play essential roles in known inflammatory pathways, the proteomics analysis also revealed a group of proteins that had been upregulated by Fh15 and downregulated by LPS that were not studied, which we consider a limitation of the present study. For example, Fh15 significantly upregulated GSS-glutathione synthetase (FC 1.676, *p*-value = 0.0415). This is a key antioxidant that protects cells from damage caused by reactive oxygen species (ROS) and other oxidative stress [82]. Fh15 also upregulated YY1 and YY2 (FC 1.572, *p*-value = 0.00214), which are transcription factors that plays roles in stemness, brain development, and potentially tumor suppression [83,84], also upregulated FADS2 (FC 1.937, *p*-value =0.000441), a key enzyme in the lipid metabolism of cells [85], NDUFV1-NADH (FC 1.581, *p*-value = 0.000689), NDUFV2-NADH (FC 2.012, *p*-value = 0.0145), NDUFA6-NADH (FC 2.396, *p*-value = 0.00236), and SDHC-subunit C (FC 1.619, *p*-value = 0.000536), which are proteins that play roles in the cell energy production, mitochondrial respiration chain and metabolism [86,87]. Although the role of many of these proteins during the sepsis is unknown it is possible to assume that their upregulation by Fh15 could in somehow contribute to the anti-inflammatory effect and contribute to the proper cell function. Therefore, more studies should be performed to validate these proteins, determine their role in sepsis and explore additional Fh15 modulatory mechanisms. Despite of this limitation, the results obtained in the present study led us to suggest that Fh15 exerts broad spectrum of action by selectively downregulating key proteins involved in interconnected signaling pathways. This multi-targeted modulation underscores Fh15′s potential as an effective biotherapeutic for sepsis treatment.

## 4. Materials and Methods

### 4.1. Animals

The study involved a total of 10 naïve inbred female or male BALB/c 6–8-week-old mice from Charles River Laboratories (Wilmington, MA, USA). Animals were euthanized by cervical dislocation under deep anesthesia and to remove the femurs and tibia, which were used for isolating bone marrow-derived cells.

### 4.2. Recombinant Fh15

cDNA encoding amino acid sequence of Fh15 (GenBank ID M95291.1) was synthesized, cloned into the pT7M vector, and expressed in *Bacillus subtilis* as a fusion protein with a 6His at the amino terminus. It was then purified by a Ni^+^-agarose column as previously described [19] and endotoxins were removed with the use of polymyxin B (PMB) columns according to the manufacturer’s instructions. Purified Fh15 was concentrated up to a maximum volume of 10 mL by AMICON Ultra Centrifugal Filters (YM-3) and its concentration adjusted to 2.29 mg/mL, as determined by the Bradford method. Western blot using a mouse anti-Histidine tag monoclonal Antibody (Cat. No. A00186, Genscript, Piscataway, NJ, USA) was used to confirm purity of the purified protein (Appendix A). Levels of endotoxin assessed by a Chromogenic Limulus Amebocyte Lysate assay (Lonza, Walkersville, MD, USA) revealed that Fh15 contained only traces at levels < 0.4 EU/mg. The purity of Fh15 was >90% as determined by densitometry analysis of the Coomassie blue stained SDS-PAGE gel under reducing conditions and confirmed by LC-MS/MS. The cDNA synthesis, cloning, expression and purification was performed in collaboration with GenScript, Piscataway NJ, USA (Order # U917RGI020).

### 4.3. Cell Line and Maintenance

In this study, RAW 264.7 cells (ATCC, Manassas, VA, USA) were used, which is a macrophage-like cell line originated from a male BALB/c mouse and transformed by the Abelson murine leukemia virus. RAW 264.7 cells were cultured on a T25 cell culture flask in Dulbecco’s modified minimal essential medium (DMEM)-high glucose with L-glutamine, sodium pyruvate and sodium bicarbonate (Sigma Aldrich, St. Louis, MO, USA) supplemented with 10% (*v*/*v*) of heat inactivated fetal bovine serum (iFBS, Sigma Aldrich, St. Louis, MO, USA) and 100 U/mL penicillin and 100 μg/mL of streptomycin (Sigma Aldrich, USA). Cells were incubated at 37 °C, 5% CO_2_ until reaching a 60–70% confluence. Then, cells were seeded at a concentration of 1 × 10^6^ cells per well in a temperature-responsive, polymer-coated 24-well plate (Nunc-Multidishes Up-Cell Surface from Thermo Fisher, Waltham, MA, USA), which allow for temperature-controlled cell detachment. Cells were then treated in triplicate with 1 µg/mL LPS-*E. coli* O111:B4 (Sigma-Aldrich, St. Louis, MO, USA), 10 μg/mL Fh15, or equivalent volume of PBS, and then incubated overnight (O/N) at 37 °C, 5% CO_2_.

### 4.4. Mouse Primary Cells Isolation and Differentiation

Bone marrow-derived macrophages (BMDMs) were isolated from the femoral and tibial shafts of naïve mice following a pre-established protocol [17]. Briefly, femoral shafts of mice were flushed with 3 mL of cold sterile PBS and the resulting cell suspension then sieved to eliminate large clumps. Cells were washed three times with sterile complete RPMI-1640 medium supplemented with 20 mM L-glutamine, 10 mM HEPES, 10% (*v*/*v*) of iFBS, 100 U/mL penicillin and 100 μg/mL of streptomycin. Cells were adjusted to 1 × 10^5^ cells per well with differentiation medium (complete RPMI-1640 supplemented with 20 ng/mL M-CSF from R&D Systems Ltd., Minneapolis, MN, USA) and cultured in 24-well NuncTM Multidishes Up-Cell Surface plate (Thermo Fisher, Waltham, MA, USA) at 37 °C, 5% CO_2_. At the third day of incubation, non-adherent cells were removed, and adherent cells were placed in fresh differentiation medium, and the incubation was prolonged for 7 days to allow full macrophage maturation, which was assessed by fluorescence-activated cell sorting (FACS) analysis and 4/80 surface antigen expression. BMDMs were seeded into 24-well plates (NuncTM Multidishes Up-Cell Surface) at 1 × 10^5^ cells per well in complete RPMI-1640 medium and then treated in triplicate with Fh15 (10 μg/mL) for 30 min before being stimulated with 100 ng/mL LPS (Fh15 + LPS) and then incubated overnight (O/N) at 37 °C, 5% CO_2_. Cells treated only with Fh15 (10 μg/mL), LPS (100 ng/mL) or PBS were used as controls.

### 4.5. ELISA Quantification of TNF-α in BMDM Culture Supernatants

After overnight incubation, the supernatants of BMDMs treated with PBS, Fh15, LPS or Fh15 + LPS were collected and used for measuring levels of secreted TNF-α using a murine TNF-α ABTS ELISA kit (Thermo Scientific, Waltham, MA, USA). The concentrations of TNF-α were estimated by using a standard curve and expressed in pg/mL in accordance with the manufacturer’s instructions.

### 4.6. Protein Extraction and Quantification

Adherent macrophage-like and BMDMs were detached, centrifuged at 16,000× *g* for 20 min, and the resulting pellets were lysed on ice at 4 °C for 30 min using cold 1X RIPA buffer containing 50 mM Tris-HCl, 150 nM NaCl, 1.0% NP-40, 0.5% sodium deoxycholate, 0.1% sodium dodecyl sulfate, and protease inhibitor cocktail (Sigma Aldrich, St. Louis, MO, USA) as described in previous studies [88,89]. Cells were centrifuged at 20,000× *g* for 10 min at 4 °C, and the supernatants transferred to clean tubes. Protein concentration was determined using a Bicinchoninic acid (BCA) assay (Thermo Scientific, Waltham, MA, USA) following the manufacturer’s instructions and stored at −80 °C until use.

### 4.7. Preparation of Protein Samples for Tandem Mass Tag (TMT) Labeling

A total of nine samples originated from macrophage-like cells (3 triplicates each of cells treated with Fh15, LPS or PBS) were transferred to new microcentrifuge tubes and the final volume adjusted to 100 μL at a concentration of 1 μg/μL and then precipitated with cold acetone. The resulting pellets were resuspended in sample buffer (95% Laemmli buffer, 5% β-mercaptoethanol) to a final concentration of 2 μg/μL. Samples were heated at 70 °C and loaded into precast TGX Mini-PROTEAN gels (Bio-Rad, Hercules, CA, USA) for short-run SDS-PAGE. Following electrophoresis, the gels were stained with Coomassie blue, and protein bands were excised. Gel pieces were destained by incubating them in a 50 mM ammonium bicarbonate and 50% acetonitrile solution at 37 °C for up to 3 h. The proteins were reduced using dithiothreitol (25 mM DTT in 50 mM ammonium bicarbonate) at 55 °C, alkylated with iodoacetamide (10 mM IAA in 50 mM ammonium bicarbonate) at room temperature in the dark, and digested with trypsin (Promega, Madison, WI, USA) overnight at 37 °C using a 1:50 trypsin-to-protein ratio for optimal digestion as performed in previous studies [88,89]. The next day, digested peptides were extracted from the gel pieces using a solution of 50% acetonitrile and 2.5% formic acid in water. The peptides were then dried and stored at −80 °C until they were ready for TMT-labeling.

### 4.8. TMT-Labeling, Fractionation, and Mass Spectrometry Analysis

TMT labeling was performed following the manufacturer’s instructions (Thermo Scientific, Waltham, MA, USA) and according to previously optimized protocols at the UPR-MSC Translational Proteomics Center [88,89]. One TMT11-plex platform was required to accommodate nine samples with their respective treatments and an additional pool of 100 μL of each of the samples before drying the extractions to regulate the abundance of peptides and normalize the volume. Triethylammonium bicarbonate (TEAB, 100 mM) buffer was used to reconstitute the dried peptides and then labeled with TMT11-plex reagents. The TMT reagents were resuspended in 41 μL of anhydrous acetonitrile (99.9%), added to each respective sample, and incubated for one-hour with occasional vortexing at room temperature as described in Zenon-Melendez et al., 2022 [90]. A representative illustration of the protocol is shown in Figure 1.

The TMT reaction was quenched with 15% hydroxylamine for 15 min. After quenching was completed, samples were subjected to fractionation. The fractionation was performed using a Pierce High pH Reversed-Phase Peptide Fractionation Kit (Thermo Fisher Scientific, Waltham, MA, USA) and following manufacturer’s instructions. Briefly, the column was equilibrated twice using 300 μL of acetonitrile, centrifuged at 5000 rpm for 2 min, and the steps were repeated using 0.1% Trifluoroacetic acid (TFA). The TMT-labeled pool was reconstituted in 300 μL of 0.1% TFA, loaded onto the column and washed to remove salt contaminants or any unbound TMT reagent. The clean-pooled sample was then eluted 8 times into 8 different vials using a series of elution solutions with different Acetonitrile/0.1% Triethylamine percentages. Fractions were then dried and reconstituted for mass spectrometry analysis using 0.1% formic acid in water (Buffer A). For peptide separation, a PicoChip H354 REPROSIL-Pur C18-AQ 3 μm 120 A (75 μm × 105 mm) chromatographic column (New Objective, Litleton, MA, USA) was used. The peptide separation was obtained using a gradient of 7–25% of 0.1% of formic acid in acetonitrile (Buffer B) for 102 min, 25–60% of Buffer B for 20 min, and 60–95% Buffer B for 6 min. Following separation, peptides were sprayed and analyzed using a Q-Exactive Plus Hybrid Quadrupole-Orbitrap (Thermo Fisher Scientific, Mont Prospect, IL, USA) operated in positive polarity mode and data-dependent mode. The full scan (MS1) was measured over the range of 375 to 1400 *m*/*z* at resolution of 70,000. The MS2 (MS/MS) analysis was configured to select the ten (10) most intense ions (Top10) for HCD fragmentation with a resolution of 35,000. A dynamic exclusion parameter was set for 30 s.

### 4.9. Protein Identification, and Bioinformatics Analyses

Protein identification was conducted according to the standardized protocol established by the Translational Proteomics Center [91]. Raw MS/MS (MS) proteomics data have been deposited to the ProteomeXchange Consortium via the PRIDE [92] partner repository with the dataset identifier PXD065520 and 10.6019/PXD065520. Raw MS/MS data files were processed using Proteome Discoverer software version 2.5 (Thermo Fisher Scientific, Mont Prospect, IL, USA) employing the SEQUEST HT algorithm. Protein identification was performed against a UniProt database specific to *Mus musculus* (mouse). The database search parameters included the following modifications: dynamic oxidation on methionine (+15.995 Da), static carbamidomethylation of cysteine (+57.021 Da), and static TMT labeling modifications on any N-terminal and lysine residues (+229.163 Da).

Statistical analysis of protein abundance was performed using R-Limma package (version 3.41.15) with Bioconductor (version 3.16) [93] following the Translational Proteomics Center pipeline [88,89]. Two case–control comparisons were conducted: LPS vs. PBS-treatment, and Fh15- vs. LPS-treatment. A fold change (FC) threshold of ≥1.5 and *p*-value ≤0.05 were used to identify significantly dysregulated proteins, with 95% confidence. This cut-off was selected to detect proteins that varied by at least 50% in relative abundance between groups. Proteins were considered upregulated when their fold change was greater than or equal to 1.5 (Log2 FC ≥ 0.585), while those with a fold change less than or equal to 0.667 (Log2 FC ≤ −0.585) were classified as downregulated. Proteins meeting either of these criteria were collectively defined as dysregulated. All significant dysregulated proteins were annotated using UniProt Accession numbers to retrieve official protein names.

To further explore functional implications, pathway enrichment analysis was performed using Ingenuity Pathway Analysis (IPA) software (version 22.0.2, QIAGEN Digital Insights, Germantown, MD, USA). The Ingenuity CORE analysis module was used to identify enriched canonical pathways, disease associations, and biological functions. Canonical pathways were considered significant at −log_10_ (*p*-value) ≥ 1.30 (equivalent to *p* ≤ 0.05). The same *p*-value threshold was applied for disease and function associations. Proteins implicated in key biological pathways—including iNOS, NF-κB, inflammasome activation, and Th1/Th2 signaling—were further validated via Western blotting. Interaction networks for dysregulated proteins in these pathways were visualized in network diagrams. All graphical representations, including bar plots, heatmaps, Venn diagrams, and pie charts, were generated using GraphPad Prism software (version 8). Statistical significance between groups was further evaluated using one-way ANOVA, followed by Tukey’s post hoc test, as implemented in GraphPad Prism.

### 4.10. Protein Validation by Quantitative Western Blotting

Protein samples generated from RAW 264.5 and BMDMs lysates were separated by means 4–20% SDS-PAGE (Mini-PROTEAN^®^ TGX™ Precast Gel, Bio-Rad Hercules, CA, USA) initially for 30 min at 60 V, followed by 1.5 h at 100 V, for a total of two rounds. Subsequently, proteins were transferred onto a polyvinylidene fluoride (PVDF) membrane (Bio-Rad, Hercules, CA, USA) at 100 V, 4 °C for 1 h using Tris-Glycine buffer (Sigma-Aldrich) with 10% of methanol. Proteins of interest were detected using polyclonal or monoclonal antibodies specific for each protein (Table 2), followed by anti-mouse or anti-rabbit secondary IgG antibody coupled to horseradish peroxidase (Cell Signaling Technology, Danver, MA, USA). Reaction was revealed by using a chemiluminescent substrate (Thermo Fisher Scientific, Mont Prospect, IL, USA). The blot images were visualized in a Chemidoc MP Imaging system (Bio Rad, Hercules, CA, USA) and densitometry analysis was performed using ImageJ software (version 1.53u) (https://imagej.nih.gov/ij/download.html, accessed on 3 November 2022). All values were expressed as fold changes over the PBS-treated cells.

### 4.11. Statistical Analysis

Data were analyzed using one-way ANOVA with multiple comparisons and the Tukey test, employing GraphPad Prism software (version 8). Differences were considered significant at a *p*-value of ≤0.05.

## 5. Conclusions

Based on the results shown in the present study, we have developed a hypothetical sequence of events in which Fh15 participates, which we believe occurs as part of the modulation mechanisms of this molecule and that could explain the dramatic reduction in the inflammatory response observed in our previous sepsis studies with animal models [18,19]. The proposed mechanism is illustrated in Figure 8 and summarized as follows.

At the beginning of a bacterial infection, TLR4 recognize LPS and along with its co-receptor MD2 forms a complex on the surface of cells that binds to LPS facilitated by CD14 [81]. Unlike Fh12, which targets CD14 to suppress TLR4/NF-κB pathway [17], Fh15 downregulates the expression of CD36 and performing it may disrupt the cooperation between CD36 and TLR4 in the pathogen-recognition, impairing bacterial phagocytosis and suppressing the production of TNF-α, and IL-1α. This suppression could help restrict the cytokine storm characteristic of sepsis. Concurrently, Fh15 suppress iNOS, thereby inhibiting classic macrophage activation (M1). This reduction limits the production of nitric oxide, reactive oxygen intermediates and superoxide radicals. Therefore, to maintain under control the levels of superoxide and limit the oxidative stress, Fh15 downregulates the excess of SOD2 produced by LPS. Concurrently, Fh15 also downregulates Lck, thus interfering with T cell activation. This modulation may favor the development of effector and memory T cells with a Th2 phenotype, which is associated with anti-inflammatory responses. Th2 cells produce anti-inflammatory cytokines like IL-4 and IL-13, which can induce macrophage to polarize to M2 phenotype. Indirectly, Fh15 could thus promote M2 macrophage polarization. All these mechanisms triggered simultaneously, could generate an anti-inflammatory environment in the cells that counterbalance the excessive inflammation typical of sepsis. However, further in vivo and in vitro studies must be performed to confirm all these putative mechanisms of action. Firstly, it would be necessary to determine if Fh15 directly binds CD36 or simply induce downregulation as observed in the present study. It is essential to perform phagocytosis assays to confirm that Fh15 suppresses the bacterial phagocytosis from macrophages and if it is coupled with reduced and sustained suppression of superoxide radicals. Moreover, the direct or indirect participation of Fh15 in the Th2/M2 polarization also should be confirmed. Moreover, the observation that Fh15 downregulates specific molecules that participate in several signaling pathways downstream TLR4 that occur in the cell cytoplasm could suggest that Fh15 penetrate the cells. However, despite Fh15 being a small molecular weight protein it cannot enter the cell by simple diffusion or pinocytosis but by endocytosis. Therefore, the putative entering of Fh15 into the cell and which mechanisms participate in this event also should be explored. Many of these studies are currently ongoing. Considering the complexity of the inflammatory response and the management of sepsis, having a drug like Fh15 that can effectively suppress inflammation by simultaneously blocking different signaling pathways would represent a huge advance in achieving a broad-spectrum therapy against the inflammation generated by sepsis.

## Figures and Tables

**Figure 1 ijms-26-06914-f001:**
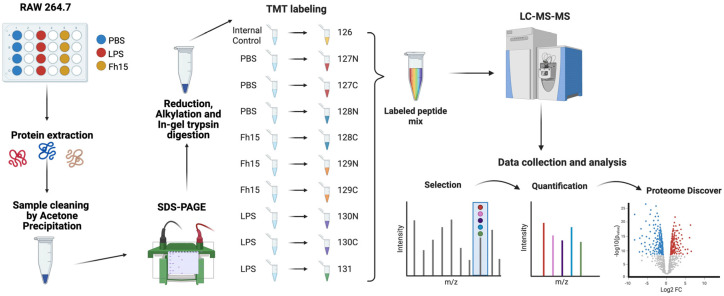
Schematic representation of the TMT-labeling protocol for the quantitative proteomics approach. RAW 264.7 cells were treated with PBS as a negative control, LPS (1 µg/mL) as a positive control and Fh15 (10 µg/mL) and then incubated for 18 h at 37 °C, 5% CO_2_. Protein extraction was performed using 1X RIPA Buffer. Acetone precipitation and SDS-PAGE were followed by reduction, alkylation, and in-gel digestion. Sample peptides were labeled, mixed, and processed using LC-MS/MS. Data was quantified and analyzed using Proteome Discoverer and Ingenuity Pathway Analysis. Created in BioRender. Armina Rodriguez, A. (2025) https://BioRender.com/5xbbgd9 (accessed on 3 July 2025).

**Figure 2 ijms-26-06914-f002:**
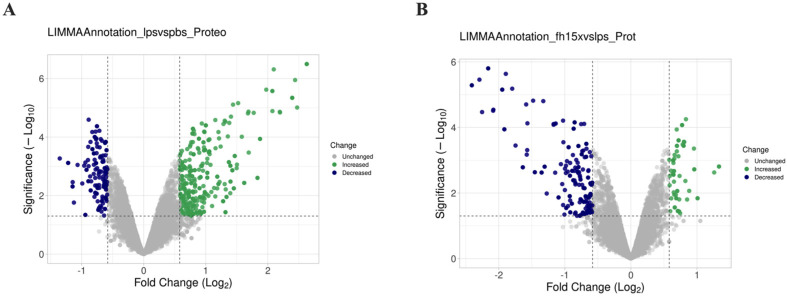
Comparative volcano plots showing differential protein expression in RAW 264.7 cells in response to LPS and Fh15. The dashed lines in the volcano plots indicate the threshold for statistical significance. The horizontal dashed line corresponds to a *p*-value ≤ 0.05 and the vertical dashed line represent a log2 fold change (FC) of ≥|0.5|. (**A**) Volcano plot comparison illustrating the distribution of proteins identified in samples treated with LPS compared to PBS-control. (**B**) Volcano plot illustrating the distribution of proteins identified in samples treated with Fh15 compared to those treated with LPS. The volcano plots were generated with VolcaNoseR [24].

**Figure 3 ijms-26-06914-f003:**
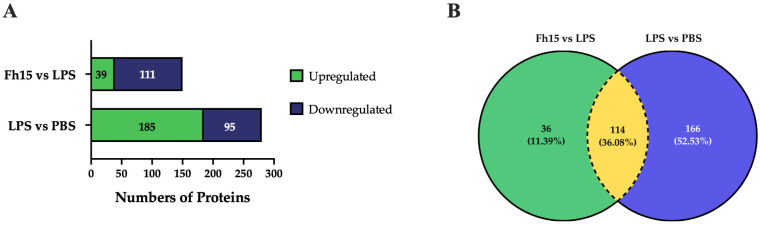
Dysregulated proteins of macrophages-like cells (RAW 264.7 cells) treated with LPS or Fh15. (**A**) Stacked bar plot depicting dysregulated proteins by the Fh15 and LPS treatments. (**B**) Venn Diagram comparing dysregulated proteins between the experimental comparisons LPS vs. PBS and Fh15 vs. LPS. Number of differentially abundant proteins commonly identified across the group’s comparisons, considering a cut-off fold change of 1.5 and *p* ≤ 0.05.

**Figure 4 ijms-26-06914-f004:**
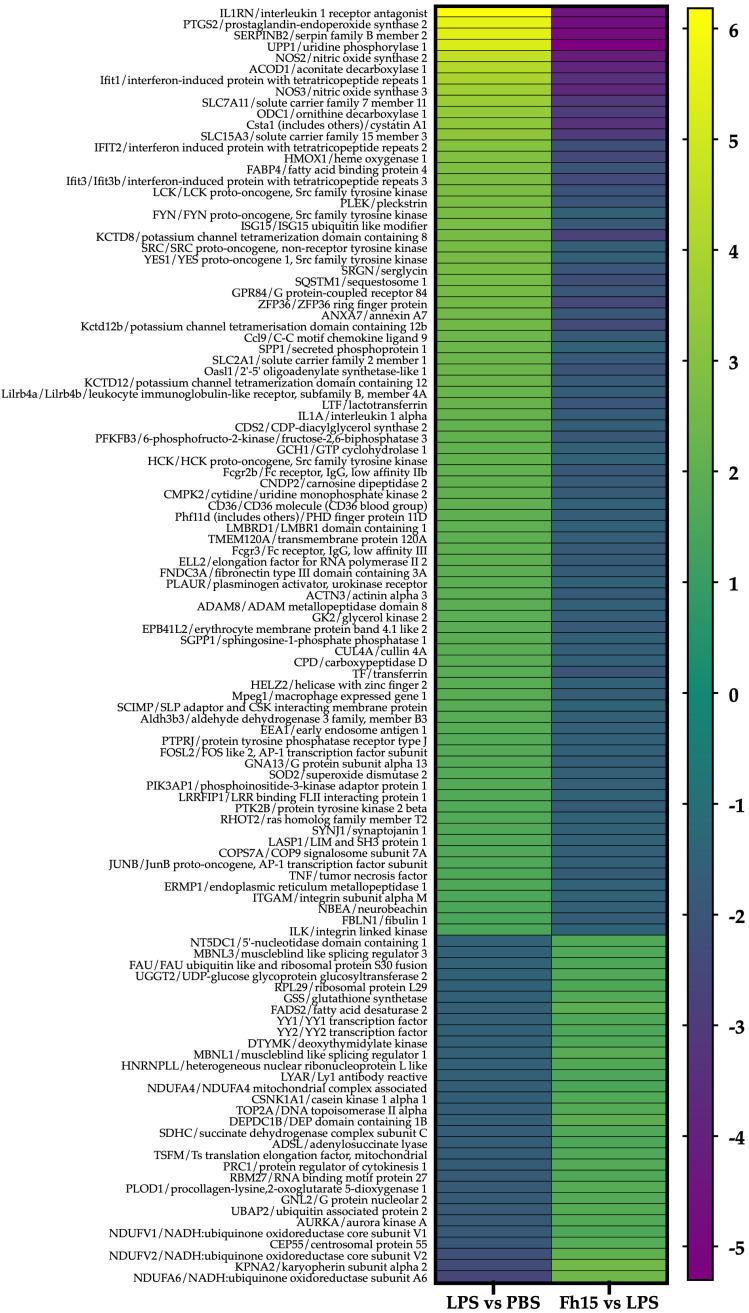
Heat maps showing fold changes in dysregulated proteins that are common to the LPS and Fh15 treatments. The figure shows the heat map of the 114 macrophages-like common proteins identified and quantified. Green-gradient color represents upregulated proteins (Fold changes (FC) ≥ 1.5, *p*-value ≤ 0.05) and blue-gradient color represent downregulated proteins (FC ≤ −1.5, *p*-value ≤ 0.05).

**Figure 5 ijms-26-06914-f005:**
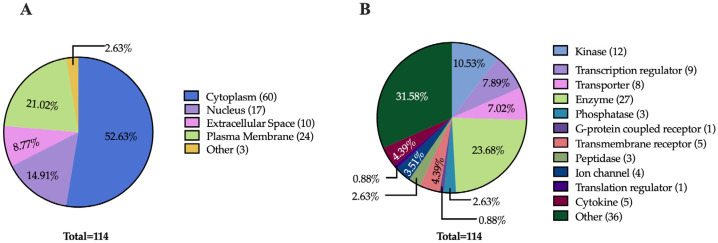
Pie Charts Depicting Subcellular and Functional Distribution of Dysregulated Proteins in Response to LPS and Fh15 Treatments. (**A**) Pie charts illustrating the proportional distribution of dysregulated proteins across subcellular compartments in the experimental comparisons LPS vs. PBS and Fh15 vs. LPS. (**B**) Pie charts showing the functional categorization of dysregulated proteins in the comparisons LPS vs. PBS and Fh15 vs. LPS.

**Figure 6 ijms-26-06914-f006:**
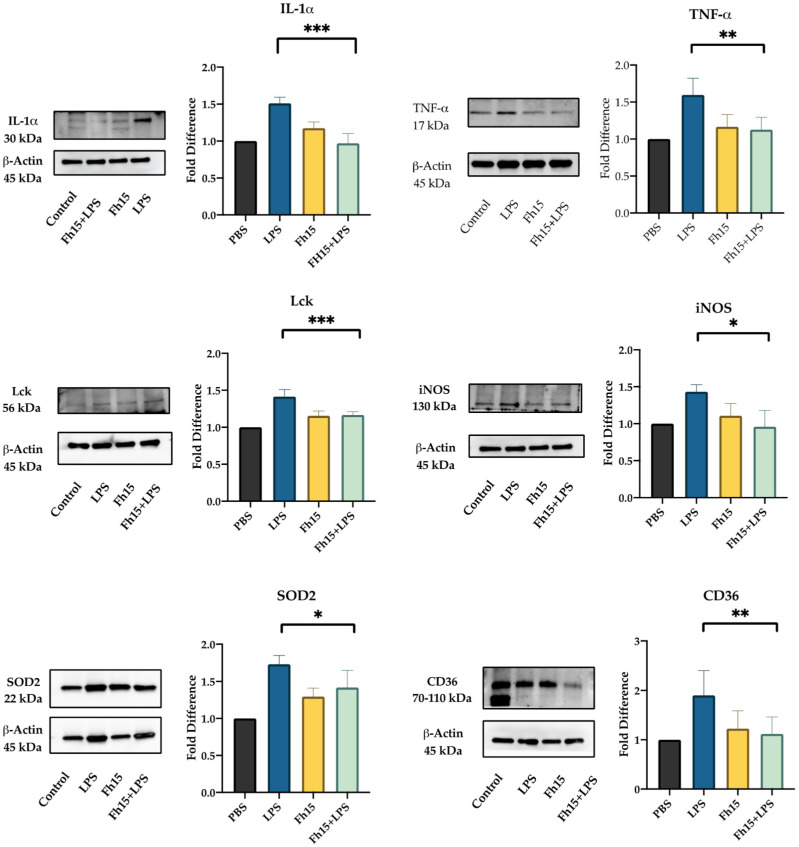
Fh15 suppresses the expression of proteins associated with several inflammatory pathways within bone marrow-derived macrophages. BMDMs were seeded into 24-well plates (NuncTM Multidishes Up-Cell Surface) at 1 × 10^5^ cells per well in complete RPMI-1640 medium and then treated in triplicate with Fh15 (10 μg/mL) for 30 min before being stimulated with 100 ng/mL LPS (Fh15 + LPS) and then incubated overnight (O/N) at 37 °C, 5% CO_2_. Cells treated only with Fh15 (10 μg/mL), LPS (100 ng/mL) or PBS were used as controls. Cells treated with Fh15 followed by LPS (Fh15 + LPS) showed a significant suppression of IL-1α (*** *p* = 0.0003), TNF-α (** *p* = 0.008), Lck (*** *p* = 0.0005), iNOS (* *p* = 0.05), SOD2 (* *p* = 0.04) and CD36 (** *p* = 0.008).

**Figure 7 ijms-26-06914-f007:**
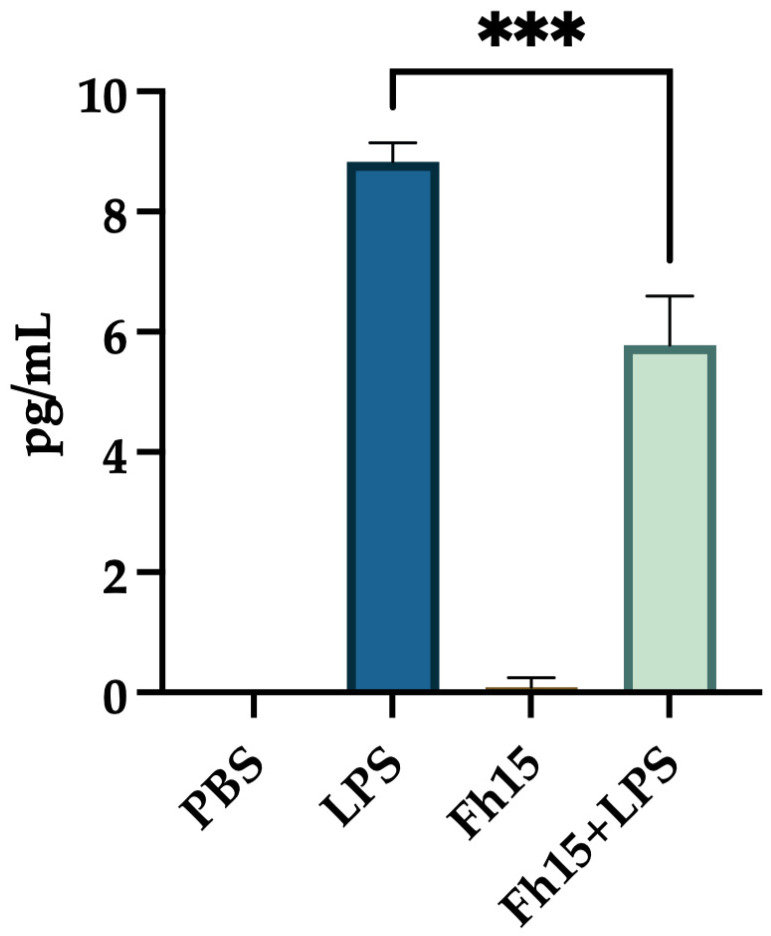
The treatment with Fh15 significantly reduced the secretion of TNF-α levels within bone marrow-derived macrophages. BMDMs (1 × 10^5^ cells) seeded into 24-well plates at 1 × 10^5^ cells per well in complete RPMI-1640 medium were treated in triplicate with Fh15 (10 μg/mL) for 30 min before being stimulated with 100 ng/mL LPS (Fh15 + LPS) and then incubated overnight (O/N) at 37 °C, 5% CO_2_. Cells treated only with Fh15 (10 μg/mL), LPS (100 ng/mL) or PBS were used as controls. Cells treated with Fh15 followed by LPS (Fh15 + LPS) significantly reduced the levels of TNF-α secreted to the culture media (*** *p* = 0.0001).

**Figure 8 ijms-26-06914-f008:**
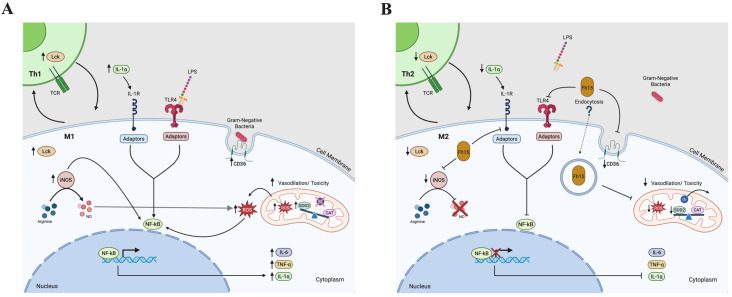
Diagram illustrating the signaling pathways leading to NF-κB activation in immune cells. The figure depicts the engagement of CD36 and TLR4 receptors during phagosome formation, triggering downstream signaling cascades that culminate in NF-κB activation. Additionally, it represents TCR activation on T cells, involving Lck, which initiates signaling events resulting in NF-κB translocation into the nucleus. These pathways collectively highlight the integration of innate and adaptive immune signals converging on NF-κB-mediated gene transcription. (**A**) Illustrate the signaling pathway activated by LPS or Gram-negative bacterial infection. (**B**) Hypothetical illustration of the inhibition caused by Fh15 in the same signaling pathways. Created in BioRender. Armina Rodriguez, A. (2025) https://BioRender.com/p48hegm; https://BioRender.com/qgkddj1 (accessed on 13 June 2025).

**Table 1 ijms-26-06914-t001:** Proteins Selected for Validation.

					Fh15 vs. LPS		LPS vs. PBS	
Symbol	Gene Name	ID	Location	Type(s)	Fold Change	*p*-Value	Fold Change	*p*-Value
*NOS2*	nitric oxide synthase 2	P29477	Cytoplasm	enzyme	−4.242	0.0000309	4.584	0.0000144
*Lck*	Lck proto-oncogene, Src family tyrosine kinase	P06240	Cytoplasm	kinase	−2.137	0.0137	2.736	0.00567
*TNF-α*	tumor necrosis factor	P06804	Extracellular Space	cytokine	−1.790	0.00493	1.597	0.00382
*IL-1α*	interleukin 1 alpha	P01582	Extracellular Space	cytokine	−1.787	0.0394	2.195	0.0055
*CD36*	CD36 molecule (CD36 blood group)	A0A0G2JFB7	Plasma Membrane	trans- membrane receptor	−1.538	0.000565	2.028	0.000765
*SOD2*	superoxide dismutase 2	P09671	Cytoplasm	enzyme	−1.665	0.000527	1.722	0.0000756

Values in green color indicate the FC of downregulated proteins, while values in red color represent the FC of upregulated proteins, in the specific comparison.

**Table 2 ijms-26-06914-t002:** Quantitative Western Blot Antibodies.

Antibody Type	Antibody Name	Company	Catalog	Clone	Dilution
*Primary*	Anti-GAPDH	Cell Signaling Technology	5174S	D16H11	1:1000
*Primary*	Anti-β-actin	Cell Signaling Technology	8457L	D6A8	1:1000
*Primary*	Anti-iNOS	Cell Signaling Technology	13120S	D6B6S	1:1000
*Primary*	Anti-IL-1α	Cell Signaling Technology	50794S	D4F3S	1:1000
*Primary*	Anti-TNF-α	Cell Signaling Technology	11948S	D2D4	1:1000
*Primary*	Anti-Lck	Cell Signaling Technology	2752SS		1:1000
*Primary*	Anti-CD36	Cell Signaling Technology	74002S		1:1000

## Data Availability

The mass spectrometry proteomics data have been deposited to the ProteomeXchange Consortium via the PRIDE [92] partner repository with the dataset identifier PXD065520 and 10.6019/PXD065520. This information was incorporated in the Methods section or just before Acknowledgements and in the Abstract, e.g., Data are available via ProteomeXchange with identifier PSD065520.

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
