# Peer review of "Quantitative Proteomics Reveals Fh15 as an Antagonist of TLR4 Downregulating the Activation of NF-κB, Inducible Nitric Oxide, Phagosome Signaling Pathways, and Oxidative Stress of LPS-Stimulated Macrophages"

_ijms, 2025, doi:10.3390/ijms26146914_

Round 1
Reviewer 1 Report
Comments and Suggestions for Authors
Fh15 has been previously shown to possess the ability to improve survival in septic shock mouse models, by suppressing LPS-induced inflammatory mediators. Thus, such protein was proposed as a TLR4 antagonist. In this scenario, Armina-Rodriguez and co-workers performed a quantitative proteomics study to characterize proteins expression in macrophages treated with LPS and Fh15, hypothesizing that Fh15 is capable of downregulating proteins downstream of the TLR4 pathway in macrophages. Thus, the authors found that 114 proteins were upregulated by LPS and then downregulated by Fh15 and, among them, TNFα, IL-1α, Lck, NOS2, SOD2, and CD36 were validated via Western blot.
I found the manuscript to be well written in English, apart from few typos, the scientific findings to be interesting and the figures to be put together nicely. Nevertheless, several issues have to be addressed prior to consider the manuscript for publication in this Journal.
Starting from the major concerns, I would recommend the authors to address the following points:
- The authors treat RAW 264.7 cells with a recombinant Fh15 they express and purify, but no data on protein purity is shown or referenced to. I would thus ask the authors to kindly show the stained gels and the LC/MS data related to Fh15.
- Which nano-UHPLC-MS system has been used for the proteomics analysis? Being this a proteomics driven paper, I think is fundamental that the authors write it clearly in the material and methods section.
- Are the raw mass spectrometry data deposited on a data repository? If yes, please insert a reference in the text. If no, could the authors provide to do so?
- This is just a suggestion. The proteomics protocol sounds too unnecessarily complicated to me. The authors lysed cells in RIPA buffer, containing Tris-HCl which interferes with TMT labeling. Then they precipitated the samples with acetone, suspended them in Laemmli sample buffer and loading them on an SDS-PAGE for a short gel run. Then proteins were in gel digested with an ammonium bicarbonate containing buffer and peptides were finally labeled. I was curious to know what’s the necessity for all of this steps. Is it for concentrating the samples? Is it for cleaning them up from tris? If so, why did the authors not selected a TMT-compatible buffer to lyse the cells with, like an Hepes or TEAB based and urea containing one? Then, samples could have been digested directly in solution, in the same buffer. This would have eliminated the possibility of sample loss during the precipitation/resuspension step, and also ameliorated samples reproducibility, especially because proteins quantification was performed prior to the precipitation and resuspension.
- Why did the authors use ammonium bicarbonate for in gel digestion prior to TMT labeling? Is true that ammonium bicarbonate is volatile, but substituting it with TEAB would have ensured the maximum possible labeling efficiency. Did the authors check for this?
- I find that BMDMs should be more clearly discussed, stressing that only they were treated first with Fh15 and then with LPS. This is clearly stated in materials and methods, but I think adding it to the results as well would benefit the understanding of the data.
- The Western Blots original files would benefit some more information, such as the sample loading order and a clear molecular weight ladder. I would thus ask the authors to add in the file such info, as well as if a particular blot was done on BMDMs or RAW 264.7 cells.
- I am sorry to state this, but some Western Blots quality is below the average standard, with the membranes being dirty and the signal too low to be clearly distinguishable from the background. I am fine with all of SOD2 blots, but I have some concerns on the other proteins ones. I am in particular referring to page 4, 5, 27, 28, 34, 37 and 38 of the original blots file. I am worried that an accurate densitometric analysis on such images is not entirely accurate. Could the authors provide better quality blots?
- Line 529 to 532: I know that the authors clearly stated that they are just drawing logical conclusion, but this part is too much of a stretch (as also the authors say at line 540). Could the authors please tone down the sentence accordingly?
As minor points, I suggest the authors to address the following ones:
- At line 520, the authors state that “Fh15 appears to primarily target CD36”, which is a misleading sentence. It seems that there is a direct interaction, which of course is not the case of the presented data. Please rephrase accordingly. Also, correct the primally
- I found Figure 1 to be very well build, except for the fact that the depicted MS is a QTRAP from Sciex, which is a low resolution mass spectrometer. I think it would be nicer if the authors substituted this icon with the one corresponding to the high resolution mass spec they used for this study, or to a generic high resolution mass spectrometer.
- Please correct typos related on CO2 in the material and methods section.
- Please correct the typo at line 405 (i.e., 105n NaCl).
- Please correct the typo at line 132.
Author Response
|
Dear Reviewer Thank you very much for taking the time to review this manuscript. Your observations and recommendations have been very useful and have greatly contributed to improve our manuscript. Please find below the detailed responses to your queries.
|
|
|
Query #1: The authors treat RAW 264.7 cells with a recombinant Fh15 they express and purify, but no data on protein purity is shown or referenced to. I would thus ask the authors to kindly show the stained gels, and the LC/MS data related to Fh15
Response. Authors acknowledge this recommendation. The appropriate reference was added to the text in the Materials and Methods section (subsection 4.2. Recombinant Fh15). Moreover, a supplementary figure (Figure S6A-B) was also provided showing the western blot and LC/MS data related to Fh15.
Query #2: Which nano-UHPLC-MS system has been used for the proteomics analysis? Being this a proteomics driven paper, I think is fundamental that the authors write it clearly in the material and methods section
Response. The instrument used for mass spectrometry analyses was the Q-Exactive Plus Hybrid Quadrupole-Orbitrap (Thermo Fisher Scientific, IL, USA) operated in positive polarity mode and data-dependent mode. This statement was included in the Materials and Methods section on the last paragraph page 14 (subsection 4.8. TMT-labelling, Fractioning and Mass Spectrometry Analysis).
Query #3: Are the raw mass spectrometry data deposited on a data repository? If yes, please insert a reference in the text. If no, could the authors provide to do so?
Response. The mass spectrometry proteomics data have been deposited to the ProteomeXchange Consortium via the PRIDE partner repository with the dataset identifier PXD065520 and 10.6019/PXD065520. This information was incorporated in the Materials and Methods section or just before the Acknowledgements and in the abstract, e.g. "Data are available via ProteomeXchange with identifier PXD065520.
Reviewer access details
Log in to the PRIDE website using the following details:
Project accession: PXD065520
Token: BVRrRYwfhOsO
Alternatively, reviewer can access the dataset by logging in to the PRIDE website using the following account details:
Username: reviewer_pxd065520@ebi.ac.uk
Password: H0xXc5Mtcrwb
We also added the following Reference to the manuscript: [92] Perez-Riverol Y, Bandla C, Kundu DJ, Kamatchinathan S, Bai J, Hewapathirana S, John NS, Prakash A, Walzer M, Wang S, Vizcaíno JA. The PRIDE database at 20 years: 2025 update. Nucleic Acids Res. 2025 Jan 6;53(D1): D543-D553. doi: 10.1093/nar/gkae1011. (PubMed ID: 39494541).
Query #4: This is just a suggestion. The proteomics protocol sounds too unnecessarily complicated to me. The authors lysed cells in RIPA buffer, containing Tris-HCl which interferes with TMT labeling. Then they precipitated the samples with acetone, suspended them in Laemmli sample buffer and loading them on an SDS-PAGE for a short gel run. Then proteins were in gel digested with an ammonium bicarbonate containing buffer and peptides were finally labeled. I was curious to know what’s the necessity for all of this steps. Is it for concentrating the samples? Is it for cleaning them up from tris? If so, why did the authors not select a TMT-compatible buffer to lyse the cells with, like an Hepes or TEAB based and urea containing one? Then, samples could have been digested directly in solution, in the same buffer. This would have eliminated the possibility of sample loss during the precipitation/resuspension step, and ameliorated samples reproducibility, especially because proteins quantification was performed prior to the precipitation and resuspension.
Response: Thank you for the recommendation. We agree with the reviewer in that the protocol performed for sample preparation prior to mass spectrometry in our Proteomics Center for this study was quite complex. The proteomics part of this study was performed several years ago (2021). We are working toward moving into in solution digestion (S-trap) and data independent acquisition (DIA) for future studies given the limitation of the Thermo Q Exactive instrumentation that is the only one available at the Center. However, the proteomics sample preparation using the described method in this manuscript has also been applied in our Proteomics Center for other studies and as in the current study, has yielded a significant number of relevant proteins from cells and tissues as demonstrated by publications in high impact journals (references below). The RIPA buffer is used by many investigators and has yielded excellent results in proteomics. This is one of the many buffers recommended by Thermo for proteomics depending upon the type of tissue sample (https://www.thermofisher.com/pr/en/home/life-science/protein-biology/protein-purification-isolation/cell-lysis-fractionation/cell-lysis-total-protein-extraction.html). The short gel run has been recommended by our collaborators in the past for sample clean-up prior to TMT and has helped us to ensure the protein received from users are in good condition (not denatured) prior to MS/MS. We also encourage the researchers to add protease inhibitors to the lysis buffer to prevent protein degradation. Several references were included and described on page 13-14 of the manuscript.
- Martínez–Matías N, Chorna N, González–Crespo S, Villanueva L, Montes–Rodríguea I, Melendez-Aponte LM, Roche–Lima A, Carrasquillo–Carrión K, Santiago-Cartagena E, Rymond BC, Babu M, Stagljar I, and Rodríguez–Medina JR. 2021. Toward the discovery of biological functions associated with the mechanosensor Mtl1p of Saccharomyces cerevisiaevia integrative multi-OMICs analysis. Scientific Reports Apr 1;11(1):7411. DOI: 10.1038/s41598-021-86671-8. PMID: 33795741 PMCID: PMC8016984
- Borges-Vélez G, Rosado-Philippi J, Cantres-Rosario YM, Carrasquillo-Carrion K, Roche-Lima A, Pérez-Vargas J, González-Martínez A, Correa-Rivas MS, and Meléndez LM. 2021. Zika virus infection of the placenta alters extracellular matrix proteome. J Mol Histol. doi: 10.1007/s10735-021-09994-w. Online ahead of print. PMID: 34264436
- Borges-Vélez G, Arroyo JA, Cantres-Rosario YM, Rodriguez de Jesus A, Roche-Lima A, Rosario-Rodriguez L, Correa-Rivas MS, Campos-Rivera M, Rosado-Philippi J, and Meléndez LM. 2022. Decreased CSTB, RAGE and Axl receptor is associated with Zika infection in the human placenta. Cells 11, 3627. https://doi.org/10.3390/cells11223627
- Rosario-Rodriguez LJ, Cantres-Rosario YM, Carrasquillo-Carrion K, Rodriguez de Jesus A, Garcia-Requeña L, Roche-Lima A, and Meléndez LM.2024. Quantitative proteomics reveal that CB2R agonist JWH-133 Downregulates NF-kappaB activation, oxidative stress, and lysosomal exocytosis from HIV infected macrophages. Int J Mol Sci. 13;25(6):3246. doi: 10.3390/ijms25063246.PMID: 38542221.
- Zenon-Melendez CN, Carrasquillo Carrion K, Cantres Rosario Y, Roche Lima A, Melendez LM. Inhibition of Cathepsin B and SAPC Secreted by HIV-Infected Macrophages Reverses Common and Unique Apoptosis Pathways. J Proteome Res. 2022;21(2):301-12. Epub 2022/01/08. doi: 10.1021/acs.jproteome.1c00187. PubMed PMID: 34994563; PubMed Central PMCID: PMCPMC9169015.
- Vélez-López O, Carrasquillo-Carrión K, Yadira M. Cantres-Rosario, Machín-Martínez E, Álvarez-Rios ME, Roche-Lima A, Tosado-Rodríguez EL, Meléndez LM. 2024. Analysis of Sigma‐1 Receptor Antagonist BD1047 Effect on Upregulating Proteins in HIV‐1‐Infected Macrophages Exposed to Cocaine Using Quantitative Proteomics. Biomedicines 12,1934. https://doi.org/ 10.3390/biomedicines12091934
- Rivera-Serrano, M; Flores-Colón, M; Valiyeva, F; Meléndez, LM; Vivas-Mejia, P. 2025. Upregulation of MMP3 promotes cisplatin resistance in ovarian cancer. Int. J. Mol. Sci. 26(9), 4012; https://doi.org/10.3390/ijms26094012
Query #5: Why did the authors use ammonium bicarbonate for in gel digestion prior to TMT labeling? Is true that ammonium bicarbonate is volatile but substituting it with TEAB would have ensured the maximum possible labeling efficiency. Did the authors check for this?
Response. While it is true that any residual ammonium bicarbonate in the samples post-drying would use up a portion of the TMT reagents, we use the TMT reagents in excess, at the amounts directed by the kit. Under these conditions we have searched our resulting MS/MS data allowing the added mass by TMT reagents to be variable/not static and we have found that we rarely identify unmodified peptides. This suggests that any residual ammonium bicarbonate is not affecting our results.
Query #6: I find that BMDMs should be more clearly discussed, stressing that only they were treated first with Fh15 and then with LPS. This is clearly stated in materials and methods, but I think adding it to the results as well would benefit the understanding of the data.
Response. Thank you for the recommendation. In the Results section (subsection 2.4. Validation of selected Downregulated Proteins by Fh15 Using Western blot) it was explicitly stated that BMDMs were first treated with Fh15 for 30 min and then stimulated overnight with LPS.
Query #7: The Western Blots original files would benefit some more information, such as the sample loading order and a clear molecular weight ladder. I would thus ask the authors to add in the file such info, as well as if a particular blot was done on BMDMs or RAW 264.7 cells
Response. Thank you for the recommendation. The original western blot files were all revised and the sample loading order as well as the molecular weight of the expected band was added. Moreover, it was clarified if the original blots provided were done on BMDMs or RAW264.7 cells.
Query #8: I am sorry to state this, but some Western Blots quality is below the average standard, with the membranes being dirty and the signal too low to be clearly distinguishable from the background. I am fine with all SOD2 blots, but I have some concerns on the other proteins ones. I am referring to page 4, 5, 27, 28, 34, 37 and 38 of the original blots file. I am worried that an accurate densitometric analysis on such images is not entirely accurate. Could the authors provide better quality blots?
Response. Thank you for this observation. We have provided as supplementary data new original membranes with improved quality. Specifically, we are providing the original blots of three replicates for each validated protein and specified which one of these replicates was used in the main text as figure-6.
Query # 9: Line 529 to 532: I know that the authors clearly stated that they are just drawing logical conclusion, but this part is too much of a stretch (as also the authors say at line 540). Could the authors please tone down the sentence accordingly?
Response. Thank you for the recommendation. We have re-written these lines as follow: “Based on the results shown in the present study, we have developed a hypothetical sequence of events in which Fh15 participates, which we believe occurs as part of the modulation mechanisms of this molecule and that could explain the dramatic reduction of the inflammatory response observed in our previous sepsis studies with animal models [18, 19]. The proposed mechanism is illustrated in Figure 7 and summarized as follows….
Query # 10: At line 520, the authors state that “Fh15 appears to primarily target CD36”, which is a misleading sentence. It seems that there is a direct interaction, which of course is not the case of the presented data. Please rephrase accordingly. Also, correct the primally
Response. Unlike Fh12, which targets CD14 to suppress TLR4/NF-κB pathway, Fh15 downregulate CD36 and doing that it may disrupt the cooperation between CD36 and TLR4 in the pathogen-recognition, impairing bacterial phagocytosis and suppressing the production of TNF-α, and IL-1α.
Query # 11: I found Figure 1 to be very well build, except for the fact that the depicted MS is a QTRAP from Sciex, which is a low-resolution mass spectrometer. I think it would be nicer if the authors substituted this icon with the one corresponding to the high-resolution mass spec they used for this study, or to a generic high resolution mass spectrometer.
Response. Thank you for this observation. The image depicted MS on Figure-1 was replaced by one of a high-resolution mass spec.
Query # 12: Please correct typos related on CO2 in the material and methods section.
Response. The typos related to CO2 were corrected
Query # 13: Please correct the typo at line 405 (i.e., 105n NaCl).
Response. The typo 105nM NaCl was corrected
Query # 14: Please correct the typo at line 132.
Response. We were unable to find the typo you are referring at line 132, could you please be more specific and describe what this typo is?
Reviewer 2 Report
Comments and Suggestions for Authors
The manuscript by Armina-Rodriguez et al. describes the mechanisms of a TLR4 antagonist, Fh15, in downregulating NF-κB, inducible nitric oxide, phagosome signaling pathways, and oxidative stress in endotoxin-stimulated macrophages. The authors utilize a quantitative proteomics approach, including tandem mass tag labeling and LC-MS/MS analysis, to identify proteins upregulated by LPS and downregulated by Fh15. TNFα, IL-1α, Lck, NOS2, SOD2, and CD36 were validated by Western blot on murine bone marrow-derived macrophages due to their relevant roles in the NF-κB, iNOS, oxidative stress, and phagosome signaling pathways. The authors conclude that Fh15 exerts a broad spectrum of action by targeting multiple downstream pathways activated by TLR4 to regulate inflammatory responses during sepsis.
This is a straightforward and technically sound proteomics study that provides new mechanistic insights into the anti-inflammatory role of Fh15. It highlights the therapeutic potential of Fh15 in managing inflammatory conditions such as sepsis. However, I have the following comments and suggestions to improve the manuscript:
- As a TLR4-mediated immunoregulator, Fh15 may function through regulating critical secreted proteins such as cytokines. Therefore, measuring components in the culture supernatant would provide a more complete understanding of Fh15’s roles. Limiting the analysis to cell lysates is a potential weakness of the current study and should be discussed.
- Including the number of proteins, alongside percentages, in the pie charts of Figure 5 is recommended for clarity.
- Catalog and clone numbers of the primary antibodies used for Western blotting should be included in Table 2.
- The original Western blot images in the Supplementary are strongly encouraged to be cropped, labeled, and better organized for clearer presentation. It is unclear which lanes correspond to which treatments, and which bands represent the target proteins. Sizes of molecular weight markers should also be indicated.
- Some Western blot data appear questionable and should be clarified by the authors. For example, based on the original image of the iNOS blot (data dated 02/07/2024 in the Supplementary), the band marked as iNOS in Figure 6 appears at a much lower molecular weight than expected for a 130 kDa protein.
Author Response
|
Dear Reviewer Thank you very much for taking the time to review this manuscript. Your observations and recommendations have been very useful and have greatly contributed to improve our manuscript. Please find below the detailed responses to your queries.
|
Query # 15: As a TLR4-mediated immunoregulator, Fh15 may function through regulating critical secreted proteins such as cytokines. Therefore, measuring components in the culture supernatant would provide a more complete understanding of Fh15’s roles. Limiting the analysis to cell lysates is a potential weakness of the current study and should be discussed.
Response. We acknowledge this recommendation. We used an ELISA kit from (Thermo Scientific, Waltham, MA, USA) available in our laboratory for measuring the levels of secreted TNF-alpha in the supernatant of BMDMs stimulated with LPS, PBS or Fh15 alone as well as in BMDMs treated with Fh15+LPS. New subsections (4.5. ELISA Quantification of TNF-a in BMDM Culture Supernatant, and subsection 2.5. Measurement of TNF-a levels in supernatant of culture from bone marrow derived macrophages) and the results were added to the main text as Figure-7, showing the significant reduction of secreted TNF-a in cells treated with Fh15+LPS. These results were also commented in the first paragraph of the Discussion section. Unfortunately, we do not have a kit from the same company or any other that would allow us to measure levels of secreted IL-1a. Despite this, we believe that having demonstrated that Fh15 reduces the levels of secreted TNF-a provides strong evidence that Fh15 exerts an important role in regulating key secreted pro-inflammation cytokines.
Query # 16: Including the number of proteins, alongside percentages, in the pie charts of Figure 5 is recommended for clarity.
Response. The number of proteins alongside percentages was included in Figure-5
Query # 17: Catalog and clone numbers of the primary antibodies used for Western blotting should be included in Table 2.
Response. This information was added to the Table 2
Query # 18: The original Western blot images in the Supplementary are strongly encouraged to be cropped, labeled, and better organized for clearer presentation. It is unclear which lanes correspond to which treatments, and which bands represent the target proteins. Sizes of molecular weight markers should also be indicated.
Response. Thank you for the recommendation. The original western blot files were all revised and better organized indicating sample loading order as well as the molecular weight of the expected band was added. Moreover, it was clarified if the original blots provided were done on BMDMs or RAW264.7 cells
Query # 19: Some Western blot data appear questionable and should be clarified by the authors. For example, based on the original image of the iNOS blot (data dated 02/07/2024 in the Supplementary), the band marked as iNOS in Figure 6 appears at a much lower molecular weight than expected for a 130 kDa protein.
Response. Thank you for this observation. We have provided as supplementary data new original membranes with improved quality. Specifically, we are providing the original blots of three replicates for each validated protein and specified which one of these replicates was used in the main text as figure-6.
Round 2
Reviewer 1 Report
Comments and Suggestions for Authors
Dear authors,
thank you for the exhaustive reply to my comments, I very much appreciated your clarity. I am glad to say that tmy major concerns are now cleared up, and that the manuscript can be accepted for publication in this Journal.
Congratulations to you all!
Reviewer 2 Report
Comments and Suggestions for Authors
Most of my concerns and suggestions have been appropriately addressed, except the updated supplementary data with new original membranes and labeled standard molecular weight was not attached.